# Examining the uptake, retention, and effectiveness of a national online type 2 diabetes self-management intervention in England (Healthy Living): A retrospective cohort study

Salwa S. Zghebi[1]*, Sarah Cotterill[2], Antonia Marsden[2], Luke Paterson[3,4], Rachel A. Elliott[3], Brian McMillan[1], Martin K. Rutter[5,6], Evangelos Kontopantelis[7,8]

1 Centre for Primary Care and Health Services Research, Division of Population Health, Health Services Research and Primary Care, School of Health Sciences, Faculty of Biology, Medicine and Health, University of Manchester, Manchester, UK, 2 Centre for Biostatistics, Division of Population Health, Health Services Research and Primary Care, School of Health Sciences, Faculty of Biology, Medicine and Health, University of Manchester, Manchester, UK, 3 Manchester Centre for Health Economics, Division of Population Health, Health Services Research and Primary Care, School of Health Sciences, Faculty of Biology, Medicine and Health, University of Manchester, Manchester, UK, 4 Health Economics Research Centre, Nuffield Department of Population Health, Medical Sciences Division, University of Oxford, Oxford, UK, 5 Diabetes, Endocrinology & Metabolism Centre, Manchester University NHS Foundation Trust, NIHR Manchester Biomedical Research Centre, Manchester Academic Health Science Centre, Manchester, UK, 6 Division of Diabetes, Endocrinology & Gastroenterology, School of Medical Sciences, Faculty of Biology, Medicine and Health, University of Manchester, Manchester, UK, 7 Division of Informatics, Imaging and Data Sciences, School of Health Sciences, Faculty of Biology, Medicine and Health, University of Manchester, Manchester, UK, 8 Division of Family Medicine, Yong Loo Lin School of Medicine, National University of Singapore, Singapore, Singapore

* salwa.zghebi@manchester.ac.uk

## Abstract

### Background

The prevalence of type 2 diabetes (T2DM) is increasing rapidly in the UK and worldwide and is linked to adverse outcomes including premature death. Previous studies have shown that developing self-management skills in this population can lead to health improvements. The National Health Service in England has implemented Healthy Living (HL), an online Diabetes Self-Management Education and Support (DSMES) intervention, offering information about T2DM and help with adopting healthy behaviours.

### Aims

To examine the uptake and retention of people living with T2DM registered with Healthy Living and how its use is associated with changes in 1-year clinical outcomes (effectiveness) compared with people with T2DM who did not register (controls).

### Methods

Anonymised linked patient-level Healthy Living and National Diabetes Audit (NDA) data were used to identify adults with T2DM in England. Multivariable logistic

which permits unrestricted use, distribution, and reproduction in any medium, provided the original author and source are credited.

**Data availability statement:** The data underlying the results presented in the study are only available through the data provider 'NHS England' and there was no DOI/accession number. All interested researchers can request to access the data through NHS England directly and authors are not permitted to share the data publicly (even as minimal underlying dataset) or make it available as per the data use agreement with NHS England. The authors did not have any special access privileges to this data. NHS England email address: england.contactus@nhs.net. Dataset names: 'Healthy Living user details' dataset, 'Healthy Living engagement' dataset, and the 'National Diabetes Audit (NDA).

**Funding:** This paper reports independent research funded by the National Institute for Health and Care Research (Policy Research Programme, HED-LINE, NIHR200933). The views expressed in this publication are those of the authors and not necessarily those of the National Institute for Health and Care Research or the Department of Health and Social Care. The funder did not have any role in the study design, data collection and analysis, decision to publish, or preparation of the manuscript. Eli Lilly provided support in the form of consulting fees for MKR, unrelated to this work, and did not have any role in the study design, data collection and analysis, decision to publish, or preparation of the manuscript. The specific role of MKR is articulated in the 'author contributions' section. EK is part-funded by the NIHR HeatlhTech Research Centre in Emergency and Acute Care (NIHR205301) and the Manchester British Heart Foundation (BHF) Centre for Research Excellence (RE/24/130017).

**Competing interests:** MKR reports receiving consulting fees from Eli Lilly and modest GSK stock ownership both unrelated to this work. He has also led research at the University of Manchester with Innovate UK and NIHR funding to independently evaluate My Diabetes My Way, a digital intervention supporting the management of diabetes. This does not alter our adherence to PLOS ONE policies on sharing data. Other authors declare no conflicts of interest.

regression models identified predictors of participation in Healthy Living (uptake). Using 1-to-5 case-control propensity score matching (on age, sex, baseline HbA1c, body mass index (BMI), blood pressure (BP), cholesterol, ethnicity, deprivation) and multivariable linear and logistic models, we examined how Healthy Living use was related to 1-year HbA1c, BMI, BP, new insulin use, and completion of eight care processes recommended for people with diabetes (effectiveness). Several sensitivity and sub-group analyses were conducted to assess the robustness of the findings.

## Results

A total of 21,820 people with T2DM activated a Healthy Living account. Compared with non-participants, account activators (cases) were more likely to be female (OR: 1.91; 95%CI: 1.85, 1.96), less likely to be Asian (OR: 0.35; 95% CI: 0.33, 0.37) or Black (OR: 0.56; 95% CI: 0.52, 0.60) compared with white people. Assessing effectiveness, 4,940 Healthy Living cases were matched to 24,685 NDA controls. Compared with controls, at 1-year, Healthy Living cases had lower HbA1c (by −1.3 mmol/mol (95% CI: −1.7, −0.8) or −0.1% (95% CI: −0.2, −0.1)); BMI (−0.2 kg/m$^2$; 95% CI: −0.3, −0.1), systolic BP (−1.2 mmHg; 95% CI: −1.6, −0.7), and diastolic BP (−0.6 mmHg; 95% CI: −0.9, −0.3); higher odds of completing care processes (OR: 1.6; 95% CI: 1.5, 1.8), but non-significant for insulin use (OR: 1.0; 95% CI: 0.8, 1.2). The results of the sensitivity and sub-group analyses were consistent with the main findings.

## Conclusions

People who activated a Healthy Living account experienced, on average, moderate health benefits compared with non-participants. These benefits would be expected to reduce risks for diabetes-related complications at a population level.

## 1. Introduction

Diabetes mellitus is one of the most common long-term noncommunicable conditions worldwide [1]. The prevalence of diabetes has increased rapidly over recent decades [2,3]. It is currently estimated that 537 million adults have diabetes, projected to increase to 643 million by 2030 and 783 million by 2045. [2] In the United Kingdom (UK), diabetes prevalence has doubled in the last 15 years and it has been described as the fastest growing health threat [4,5], currently affecting >5.8 million people, of which nearly 4.6 million people live with diabetes diagnoses [6]. Direct diabetes care costs in the UK were £10.7 billion for 2021/22, accounting for 6.3% of the total National Health Service (NHS) budget in 2023/24, with projections to increase to £17.9 billion over 15 years [7]. Type 2 diabetes (T2DM) is the most common form, affecting 90–95% of all people with diabetes with a prevalence of 7% [8,9]. T2DM is associated with microvascular and macrovascular complications including blindness, amputations, and cardiovascular disease (CVD), and increased risk of premature death compared with people without diabetes [10,11].

Previous studies have shown improvements in health outcomes in people with diabetes who are equipped with self-management skills or engaged in education programmes [12–15]. Several structured diabetes self-management education and support (DSMES) programmes exist, which provide guidance on making changes, e.g., improving diet and physical activity, medication adherence, and learning to cope with negative emotions [16]. For example, the Diabetes Education and Self-Management for Ongoing and Newly Diagnosed (DESMOND), that is recommended by the UK's National Institute for Health and Care Excellence (NICE) [17–19], reported improvements in 1-year body weight loss outcome, but no evidence of a difference to glycated haemoglobin (HbA1c) levels [19]. Some systematic reviews reported the effectiveness of digital DSMES in improving HbA1c in people with T2DM by −0.6% at 12 months [20,21]. Digital interventions offer a suitable alternative by overcoming barriers often faced by participants of in-person sessions, for example, work, caring responsibilities, costs, or comorbidities [22]. It is recommended that effective management of long-term conditions should also consider reducing health inequalities, as interventions that are more effective in advantaged populations may increase disparity [23].

HeLP-Diabetes was shown to be a cost-effective web-based digital DSMES programme for people with T2DM where trial participants achieved lower 1-year HbA1c levels compared with usual care (control group) (−0.24%; 95% CI: −0.44, -0.049) [24–26]. In 2019, NHS England commissioned the development and national rollout of *HeLP-Diabetes*, named *Healthy Living for People with type 2 diabetes* (*Healthy Living*). Unlike HeLP-Diabetes, which was an unstructured programme with facilitated access via general practice and included a moderated discussion forum, Healthy Living is a self-contained and self-directed online programme. The Healthy Living website is composed of 895 web pages containing videos, articles, tools, and self-assessment quizzes [27]. The content is broadly divided into three main sections: 1) the 'Learn journey' section: a structured curriculum comprised of different educational modules, 2) the 'Find answers' section: unstructured content section dedicated to different type 2 diabetes-related topics, and 3) the 'Tools' section: includes a range of *Goals* and *Tracker* tools based on the HeLP-Diabetes website where service users can set goals and self-monitor their health. More information on the intervention and its development timeline has been reported previously from this programme of work [27,28].

Given that the NHS is investing in implementing Healthy Living across England after the HbA1c reduction observed in HeLP-Diabetes, it was important to examine whether the national-level roll-out of Healthy Living shows similar benefits. In this study, we aimed to examine if there was variation in uptake and retention of participants with T2DM in the Healthy Living programme in England, by age, sex, and other characteristics. We also examined the programme's effectiveness across several clinical outcomes at 1-year, compared with people with T2DM who did not take part in the programme.

## 2. Methods

### 2.1. Data sources

Patient-level data from three linked datasets were used in the analyses: 'Healthy Living user details' dataset, 'Healthy Living engagement' dataset, and the 'National Diabetes Audit (NDA)'. The datasets were linked using pseudonymised NHS number or the Healthy Living user identifier. The 'Healthy Living user details' dataset included data on the baseline demographics of people with T2DM registered with Healthy Living up to September 2023 in the evaluation period. The 'Healthy Living engagement' dataset included data on the engagement and usage activities of participants with Healthy Living educational modules and content. The NDA, which covers most of the English population with T2DM [29], provided data on the audit years 2020−2021, 2021−2022 and 2022−2023, including demographics (age, sex, ethnicity, social deprivation level), diabetes diagnosis date, clinical data (HbA1c, cholesterol, blood pressure, comorbidities), body mass index (BMI), and medications. Each audit year spanned 15 months between January and March the following year. To account for duplicate entries of measures recorded during the annually overlapping January to March period between each two consecutive audit years, we deduplicated these measures by using only one measure per participant per calendar year (e.g., an HbA1c measure dated February 2021 would appear twice, in years 2020−21 and 2021−22).

## 2.2. Study population

People with T2DM aged ≥18 years in England from the Healthy Living user dataset and the NDA were eligible for inclusion in the study cohort. Participants with missing pseudonymised NHS number were excluded (9% of the total cohort). Based on definitions described in Table 1, the study population was categorised into five study groups: Not registered with Healthy Living (Group 1); Registered with Healthy Living (Group 2); Activated a Healthy Living account (Group 3); Attended Healthy Living (Group 4); and Completed Healthy Living programme (Group 5).

## 2.3. Ethics statement

This study was reviewed and approved by the Yorkshire & The Humber – Leeds West Research Ethics Committee on 18 August 2020 (reference number: 20/YH/0250). The research team undertook secondary analysis of pseudonymised datasets and were not responsible for data collection or obtaining informed consent.

## 2.4. Data analysis

### 2.4.1. Uptake and retention of Healthy Living.
Descriptive analysis and regression modelling were used to analyse socio-demographic and health characteristics of participants in Healthy Living, compared with NDA controls who did not participate. These analyses aimed to provide insights into differences in the characteristics of people who participated at various stages and allowed us to understand whether there was variation in participation. The outcomes of interest were binary measures of programme participation and attendance (described below).

Continuous variables were described as mean and standard deviation (SD) or, if non-normally distributed, as median and interquartile range (IQR). Categorical variables were described as frequencies and percentages. Baseline socio-demographic, clinical measures and treatments reported for each of five study groups were: age, sex, ethnicity (White, Black, Asian, Mixed, Other), smoking status (current smoker, ex-smoker, non-smoker, never smoker), index of multiple deprivation (IMD) quintiles (IMD Q1 denotes most deprived areas, IMD Q5 for least deprived areas), Healthy Living referral route, HbA1c, BMI, systolic and diastolic blood pressure, total cholesterol, serum creatinine, T2DM duration, comorbidities (ischemic heart disease (IHD), history of CVD admission, learning disability (LD), and severe mental illness (SMI)), and baseline prescriptions of antihypertensive drugs, insulin, non-insulin diabetes drugs, and statins.

**Table 1. Definitions used to define the study population groups.**

- **Registered with Healthy Living (HL) (Group 2):** To gain access to the HL website, participants registered online giving their type 2 diabetes status, name, email address, gender, date of birth, postcode, and how they heard about HL.
- **Activated HL account (Activators) (Group 3):** Following registration, participants were sent an email with a link to complete their profile and activate their account. Participants were then able to access the intervention or programme content.
- **Attended/attendance (Group 4):** if an engagement (completed activation and completed the first module of the Learn Journey, "Introduction to type 2 diabetes") is recorded for a user within 9 months (274 days, inclusive) of account creation date.
- **Completed/completion (Group 5):** if a user accessed the 60% trigger point within the Learn Journey within 9 months (274 days, inclusive) of account creation date.
- **Uptake:** the extent to which people with T2DM start Healthy Living (HL), defined as account creation.
- **Retention:** refers to the extent people continue to use HL, assessed as the level and measures of engagement and participation in the HL programme, e.g., registered vs. attended and registered vs. completed.
- **Learn Journey:** The structured part of the HL programme comprising one-page articles grouped into sections. Participants are required to work through them in a defined order (i.e., Section 2 only becomes available after the participant has completed Section 1, etc.) The majority of the Learn Journey is the same for all participants, but there is some variation depending on whether participants opt to receive information on smoking, driving and working.

Using multivariable logistic regression models, we estimated adjusted odds ratios (OR) and 95% confidence intervals (95% CI) to examine what factors are related to participation in the programme at every stage. We evaluated the association between the aforementioned baseline measures and various measures of *participation* by fitting regression models for the following paired groups: not registered (Group 1) vs. Healthy Living account activators (Group 3); not registered (Group 1) vs. Healthy Living attendees (Group 4); not registered (Group 1) vs. Healthy Living completers (Group 5); Group 3 vs. Group 4; and Group 3 vs. Group 5.

We also evaluated the association between baseline measures and two measures of *engagement:* firstly, the total time spent on the programme website (dependent variable modelled as ≤ median time vs. > median); and secondly, the number of sessions visited on the website throughout the programme (dependent variable modelled as 1 session vs. > 1 sessions). A session was defined as a string of continuous activity on the website, ending when the user was inactive for over 10 minutes. Sub-group analyses were as described above but categorized by the timing of Healthy Living account activation as participants activating an account up to 31/08/2021 are part of the development and testing phase (Phase 1), while participants activating an account from 01/09/2021 are part of the finalised version (Phase 2).

**2.4.2. Effectiveness of Healthy Living.** Using a matched cohort study design, these analyses were based on Healthy Living cases with T2DM registered for Healthy Living up to 31st March 2022, matched to NDA controls. The 1-year outcomes were HbA1c (primary outcome), BMI, systolic blood pressure (SBP), diastolic blood pressure (DBP), and two binary outcomes: eight diabetes care processes, and new insulin use. Care processes were utilised as a marker for the quality of a patient's diabetes care, as general practices offer people with T2DM a standardised set of annual care processes to help in early identification of diabetes complications [30]. In the UK, the eight care processes include HbA1c, blood pressure, BMI, foot risk surveillance, serum cholesterol, serum creatinine, smoking status, and urine albumin/creatinine ratio.

Descriptive statistics for all baseline measures of the matched cohorts were described as above. We employed a sequential 1:5 case-control propensity score matching, to adjust for pre-treatment observable differences between a group of cases (Healthy Living registrants) and a group of untreated participants (NDA controls not registered with Healthy Living programme). Cases and controls were matched on age, sex, baseline HbA1c, BMI, SBP, total cholesterol, ethnicity, and IMD quintile using the -*psmatch2*- Stata command with a caliper size of 0.002. Multiple imputation using the *mi suite* of commands in Stata was used to impute missing values of baseline variables and outcomes (10 imputations). We included all covariates, including outcomes (effectiveness analyses), regardless of missingness level [31]. Matching was based on data imputed in the first imputation (m = 1) out of 10 imputations.

Multivariable linear and logistic regression models were fitted using the -*mi estimate*- Stata command with robust standard errors to account for clustering by case-control matching identifiers. Unless otherwise stated, all multivariable models were adjusted for the binary case-control status variable, age, sex (vs. reference category: male), ethnicity (vs. reference: White), IMD quintiles (vs. reference: most deprived), smoking status (vs. reference: never smoked), BMI, and T2DM duration; baseline IHD (vs. reference: unknown), history of CVD admission (vs. reference: unknown), LD (vs. reference: unknown), and SMI (vs. reference: diagnosis not provided); baseline antihypertensive drugs, insulin, non-insulin diabetes drugs, and statins. To proxy for diabetes severity, the insulin use outcome was defined as new insulin use in participants who were not prescribed insulin at baseline.

Several sensitivity analyses were conducted to test the robustness of the main analysis: i) excluded variables that are imbalanced by 5% and 10% between matched cases and controls; ii) regression models limited to participants with complete data, i.e., as a complete case analysis (CCA); iii) matched cohorts and analyses based on different imputations (m = 5 and m = 9). Sub-group analyses were also conducted and included: i) explore sub-groups that potentially benefit the most or the least from the intervention, by modelling the relationship between the intervention and a population characteristic (age, sex, ethnicity, IMD) as an interaction term in the regression models; ii) analyses limited to Healthy Living attendees and completers; iii) explore the association between outcomes and key general practice-level characteristics: list size,

proportion of people with T2DM, deprivation, and urban/rural classification; iv) explore effectiveness of Healthy Living by the timing of Healthy Living account activation (Phases 1 and 2). In accordance with mandatory data provider Statistical Disclosure Control (SDC) rules (such as, rounding and small number suppression), individual categories may not sum to the total, and percentages may not sum to 100% in the reported results. All analyses were performed using Stata Statistical Software, Release 18 [32].

## 3. Results

### 3.1. Uptake and retention of Healthy Living

A total of N = 21,820 people activated a Healthy Living account between May 2020 and March 2022, N = 8,375 (38%) attended, while N = 990 (5%) people completed the Healthy Living programme (Fig 1, S1 Table).

Assessing uptake (Fig 2), people who activated an account, compared with NDA controls, were more likely to be female (OR: 1.91; 95% CI: 1.85–1.96), or from least deprived areas (OR: 2.09; 95% CI: 2.00–2.19) compared with most deprived areas, or to be prescribed non-insulin diabetes medication (OR: 2.17; 95% CI: 2.10–2.25), but less likely to be Asian (OR: 0.35; 95% CI: 0.33–0.37) or Black (OR: 0.56; 95% CI: 0.52–0.60) compared with White people, or current smokers (OR: 0.49; 95% CI: 0.47–0.52) compared with never smokers (Fig 2). Some participants met more >1 exclusion criterion; therefore, numbers may not sum exactly.

Assessing retention, people who attended Healthy Living, compared with those who activated an account but did not attend, were on average more likely to be female (OR: 1.35; 95% CI: 1.27–1.44) (S1 Fig), from least deprived areas compared with most deprived areas (OR: 1.11; 95% CI: 1.02–1.22), but less likely to be of Asian (OR: 0.53; 95% CI: 0.48–0.60) or Black ethnicity (OR: 0.57; 95% CI: 0.49–0.67) compared with White ethnicity, or to be prescribed non-insulin diabetes medication (OR: 0.79; 95% CI: 0.73–0.85) (S1 Fig). Similar results were also found when comparing Healthy Living completers with those who activated an account but did not complete, despite a relatively small number of completers.

Among account activators with available engagement data, the median time spent on Healthy Living website was 6.9 mins (IQR 0.4–26.3). We compared longer attendees, who attended for more than the median time (N = 10,710), and shorter attendees, who attended for the median or less (N = 10,730). Longer attendees were, on average less likely to be male or to be of Asian or Black ethnicity, less likely to be prescribed insulin or non-insulin diabetes medication or statins, and their time since T2DM diagnosis was lower than shorter attendees (S2 Table). People who attended multiple sessions, were on average older, less likely to be male or to be of Asian ethnicity, to be from the most deprived areas, prescribed non-insulin diabetes medication or statins, and their time since T2DM diagnosis was shorter when compared with those who attended one session (S3 Table).

### 3.2. Effectiveness of Healthy Living

Overall, N = 4,940 Healthy Living account activators were matched to N = 24,685 controls; N = 1,660 attendees matched to N = 8,290 controls; and N = 245 completers matched to N = 1,215 controls. Matching resulted in balanced baseline characteristics for most variables, except some, e.g., smoking status, history of CVD, and completion of care processes (S4 and S5 Tables).

After adjusting for available baseline characteristics and health measures, participants who activated a Healthy Living account had lower HbA1c at 1 year (primary outcome), on average, than those in a matched NDA control group, by −1.3 mmol/mol (95% CI: −1.7, −0.8) or −0.1% (95% CI: −0.2, −0.1) in Diabetes Control and Complications Trial (DCCT) units (S6 Table, Fig 3). After similar adjustments, participants who activated a Healthy Living account had lower values for secondary outcomes, BMI (−0.2 kg/m², 95% CI: −0.3, −0.1), systolic blood pressure (−1.2 mmHg; 95% CI: −1.6, −0.7), and diastolic blood pressure (−0.6 mmHg; 95% CI: −0.9, −0.3) at 1 year, compared with control, and higher odds of completing diabetes eight care processes (OR: 1.6; 95% CI: 1.5, 1.8) compared with controls (Table 2).

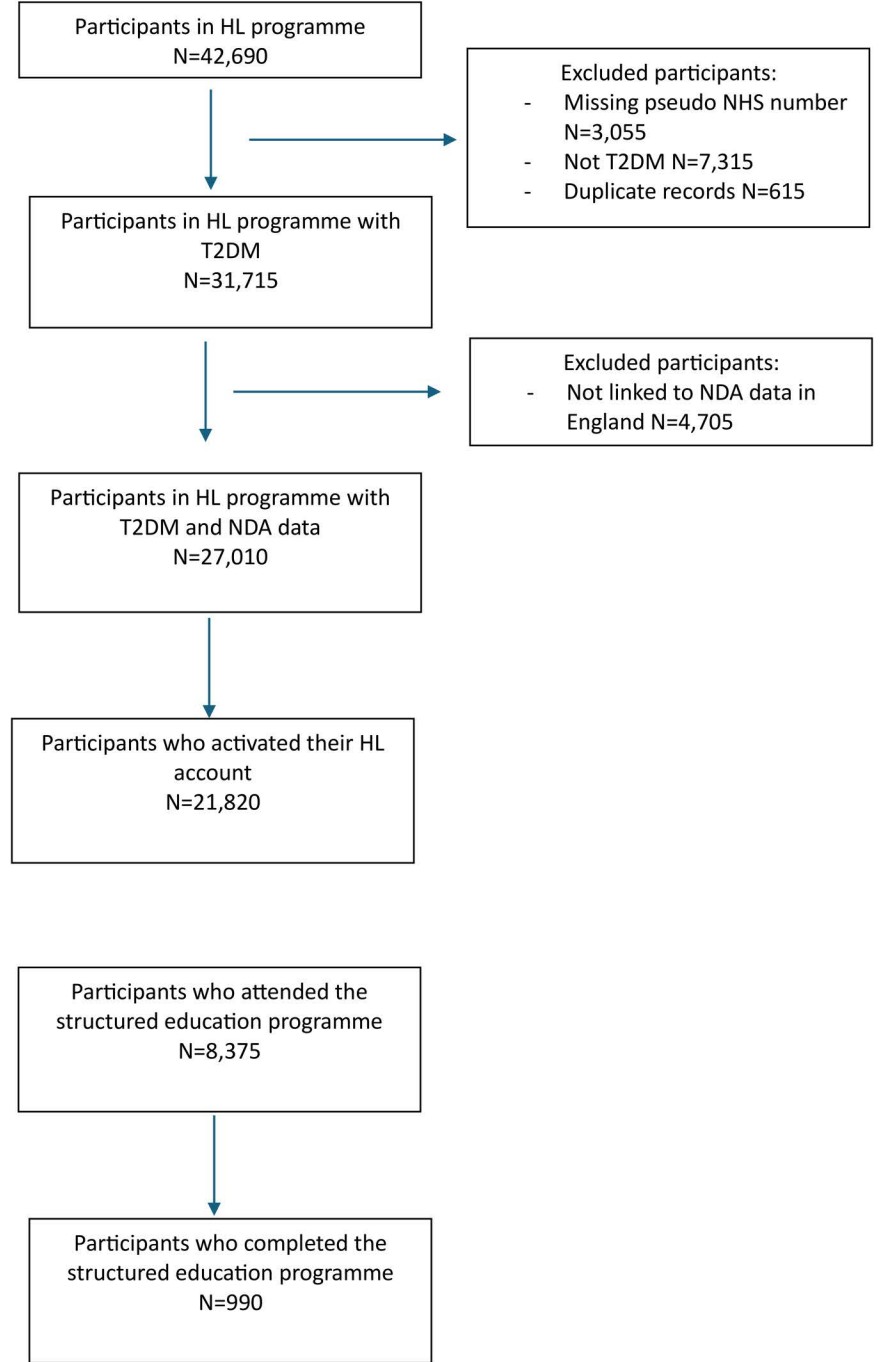

**Fig 1. Flow diagram of the included cohort.**

The results of all sensitivity analyses were consistent with the main findings (S7-S9 Tables), except that effect was not statistically significant in the CCA based on a small cohort. In the first sub-group analysis, all interaction terms were not statistically significant, suggesting no evidence of a difference in effect size between any sub-groups, apart from the case-sex interaction that was statistically significant, indicating that the effect on 1-year HbA1c change in men was greater than

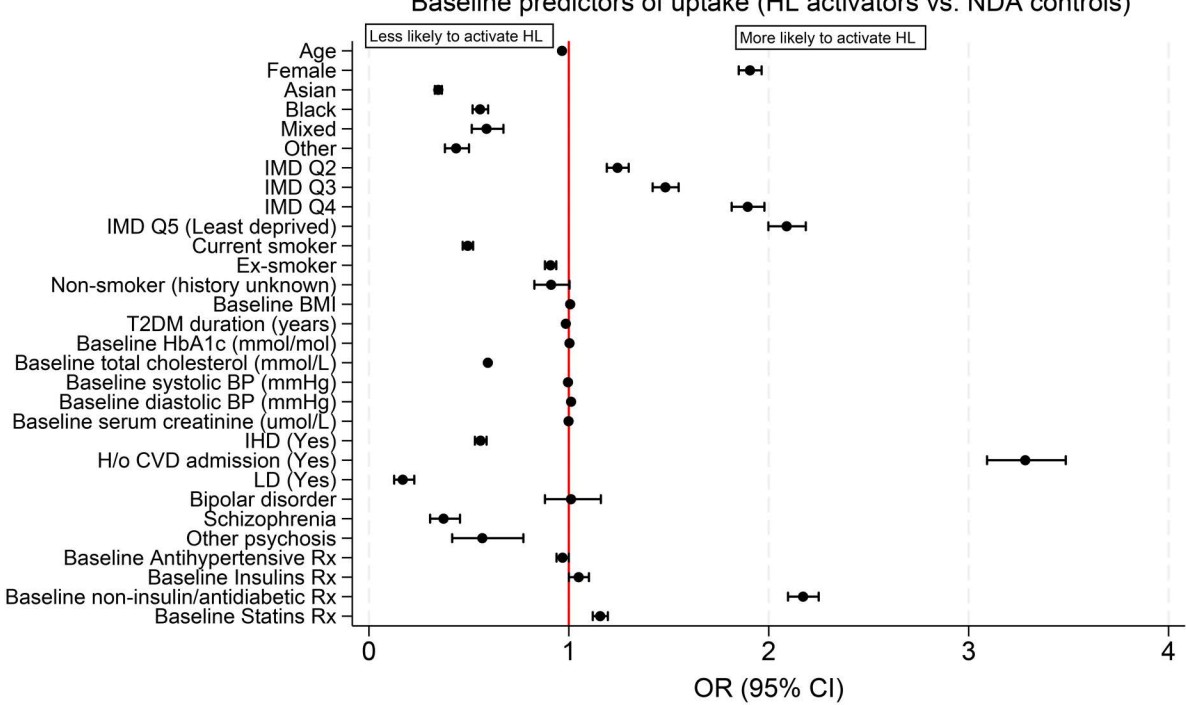

**Fig 2. Baseline predictors of Healthy Living (HL) uptake (activators) in comparison to NDA controls (odds ratios OR, 95%CI).** BMI: body mass index; CVD: cardiovascular disease; DBP: diastolic blood pressure; HbA1c: glycated haemoglobin; HL: Healthy Living; H/o: history of; IHD: ischaemic heart disease; IMD Q: index of multiple deprivation quintile; LD: learning disability; NDA: National Diabetes audit; OR: odds ratio; Rx: prescription; SBP: systolic blood pressure; T2DM: type 2 diabetes. Models were adjusted for: age, sex (reference category: male), ethnicity (reference category: White), IMD quintiles (reference category: most deprived), smoking status (reference category: never smoked), BMI, and T2DM duration; baseline ischemic heart disease (reference category: unknown), history of cardiovascular disease admission (reference category: unknown), learning disability (reference category: unknown), and severe mental illness (reference category: diagnosis not provided); baseline prescriptions of antihypertensives, insulin, non-insulin diabetes medications, and statins.

among women. In the second sub-group analysis, Healthy Living attendees had lower 1-year HbA1c levels than matched NDA controls by −2.3 mmol/mol (95% CI: −3.0, −1.6) and completers by −4.0 mmol/mol (95% CI: −6.1, −1.9) (S10 Table). We found no evidence of an association between the 1-year HbA1c and practice-level characteristics (third sub-group analysis). The effect of activating a Healthy Living account, compared with NDA matched controls was similar for all outcomes, regardless of whether the account was activated during the development/testing or finalised phases (fourth sub-group analysis, S11 Table).

## 4. Discussion

### 4.1. Summary of the findings

In this study, we included a large cohort of over 3 million people with T2DM in England, of which a small proportion (0.7%) registered with Healthy Living programme or activated an account. The main predictors of activating a Healthy Living account (uptake) included being female sex, from least deprived areas, and White ethnicity. We observed that, compared with controls, people who activated their Healthy Living account had statistically significant reductions in health outcomes, except new insulin use, including lower 1-year levels of HbA1c (−1.3 mmol/mol); BMI (−0.2 kg/m$^2$), systolic BP (−1.2 mmHg), and diastolic BP (−0.6 mmHg); and had a 1.6-fold higher odds of completing eight care processes. These benefits on outcomes were greater in those who attended or completed the programme i.e., participants with higher

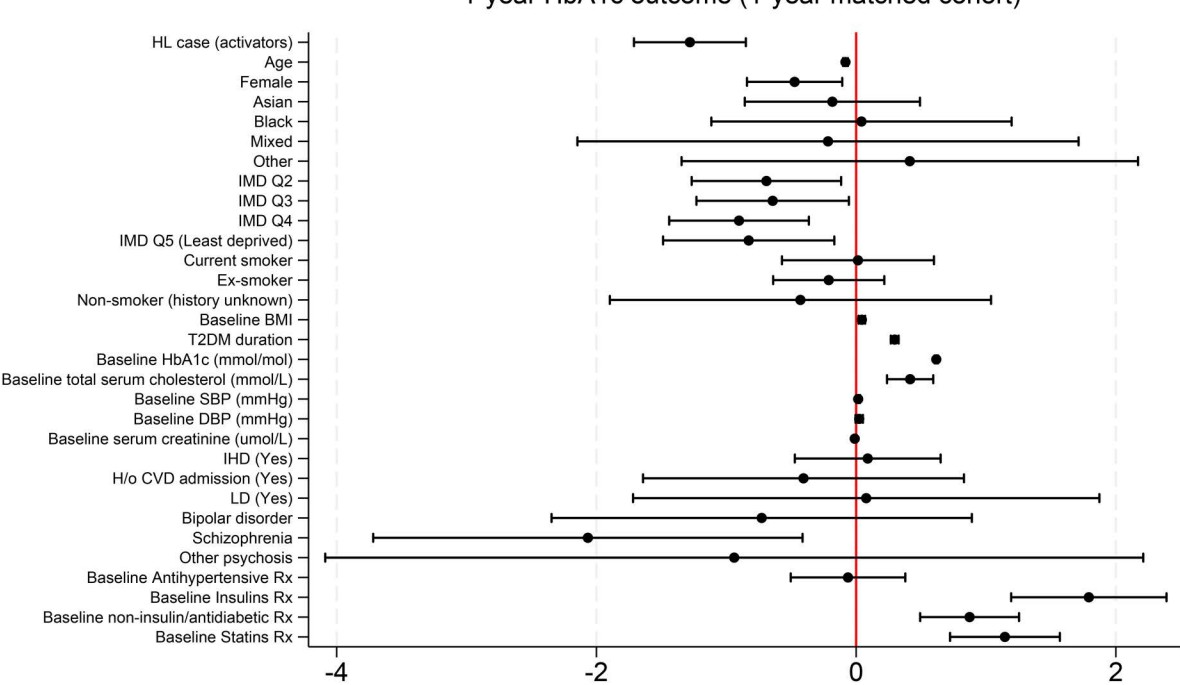

**Fig 3. Covariate-adjusted 1-year HbA1c beta coefficients (95% CI) in Healthy Living (HL) account activators compared with NDA controls.**
BMI: body mass index; CVD: cardiovascular disease; DBP: diastolic blood pressure; HbA1c: glycated haemoglobin; HL: Healthy Living; H/o: history of; IHD: ischaemic heart disease; IMD Q: index of multiple deprivation quintile; LD: learning disability; NDA: National Diabetes audit; OR: odds ratio; Rx: prescription; SBP: systolic blood pressure; T2DM: type 2 diabetes. Models were adjusted for: age, sex (reference category: male), ethnicity (reference category: White), IMD quintiles (reference category: most deprived), smoking status (reference category: never smoked), BMI, and T2DM duration; baseline ischemic heart disease (reference category: unknown), history of cardiovascular disease admission (reference category: unknown), learning disability (reference category: unknown), and severe mental illness (reference category: diagnosis not provided); baseline prescriptions of antihypertensives, insulin, non-insulin diabetes medications, and statins.

engagement. The sensitivity and sub-group analyses showed overall similar results to those from the main analysis, however, with caveats of limited power due to the smaller cohorts included in some of these analyses.

### 4.2. Findings in light of current literature

Unlike in clinical trials that are mainly based on pre-selected population, we present a real-world population-based assessment of an online free universally accessible DSMES programme, while assessing a range of clinical outcomes.

In comparison with HeLP-Diabetes [25], participants who activated an account with Healthy Living observed a smaller reduction in HbA1c level at 1 year (HL mean difference: −1.3 mmol/mol; 95% CI: -1.8, −0.9 vs. HeLP-Diabetes −2.6; 95%CI: −4.8, −0.5) (S12 Table). Healthy Living also showed a smaller effect on SBP at 1 year (mean difference: Healthy Living −1.2 mmHg vs. HeLP-Diabetes −3.8 mmHg). Healthy Living was associated with reduced BMI (mean difference: −0.2 kg/m²), reduced diastolic blood pressure (mean difference: −0.9 mmHg), and greater completion of the eight care processes (OR: 1.6), unlike in the HeLP-Diabetes trial which found no evidence of a difference on these outcomes. Importantly, because of the larger sample size in our analyses than the HeLP-Diabetes trial, there was less uncertainty around our estimates (S12 Table). In comparison with other DSMES interventions, a meta-analysis of digital DSMES interventions in adults with T2DM from eight randomised trials reported a mean difference of −0.6% (−1.0, −0.2) in 1-year HbA1c [20]. These eight trials included a total of 657 participants with T2DM in the intervention arm and 587 participants in the control

**Table 2. Multivariable-adjusted difference (95% CI) in 1-year outcomes in Healthy Living (HL) account activators compared with NDA controls (N = 29,625).**

|  | Differences (95% CI) in outcomes |
| --- | --- |
| **Linear regression models (Beta coefficient, 95% CI)** | |
| HbA1c (mmol/mol) | −1.3 (−1.7; −0.8) |
| HbA1c (%) | −0.1 (−0.2; −0.1) |
| Body mass index (BMI) (kg/m²) | −0.2 (−0.3; −0.1) |
| Systolic blood pressure (SBP) (mmHg) | −1.2 (−1.6; −0.7) |
| Diastolic blood pressure (DBP) (mmHg) | −0.6 (−0.9; −0.3) |
| **Logistic regression models (OR, 95% CI)** | |
| Insulin use | 1.0 (0.8; 1.2) |
| Completion of eight care processes | 1.6 (1.5; 1.8) |

HbA1c: glycated haemoglobin; HL: Healthy Living; NDA: National Diabetes audit; OR: odds ratio.

Models were adjusted for: age, sex (reference category: male), ethnicity (reference category: White), IMD quintiles (reference category: most deprived), smoking status (reference category: never smoked), BMI, and T2DM duration; baseline ischemic heart disease (reference category: unknown), history of cardiovascular disease admission (reference category: unknown), learning disability (reference category: unknown), and severe mental illness (reference category: diagnosis not provided); baseline prescriptions of antihypertensives, insulin, non-insulin antidiabetic medications, and statins.

arm, while assessing different digital interventions, such as web-based, mobile text and/or video messages. Additionally, they reported that interventions that offered multiple digital routes (e.g., mobile, social media) and/or multiple digital tools (e.g., text messages) showed better impact on glycaemic control than other tools. The smaller effects in Healthy Living of −0.1% (−0.2, −0.1) compared to those trials' digital DSMES interventions (−0.6%) maybe driven by greater clinical input, personalisation and resource use, real-time phone/text support, and/or reminders for self-care activities in those interventions.

An HbA1c reduction of 3–5 mmol/mol [33,34] is considered to be clinically significant for reducing the risk of diabetes-related complications at the individual patient level [33]. Although the improvement in HbA1c in our study (1.3 mmol/mol) is smaller, such modest effects can be common in real-world evaluations where patient characteristics, varied engagement and attitudes towards glucose control, and routine practice conditions may affect the magnitude of observed change compared to controlled trials. Nonetheless, this small average reduction can potentially provide modest benefits at population level for reducing the risk of developing long-term diabetes-related microvascular and macrovascular complications.

Overall, the observed variation in Healthy Living uptake aligns with evidence from other chronic condition self-management programmes [23]. In the UK, retention in the myDESMOND digital DSMES was lower in people aged <50 years, while engagement was lower in people aged <50 years, males, and non-White users [35]. myDESMOND is a digital DSMES programme for ongoing and newly diagnosed diabetes with 21,285 users. DSMES services need to address the challenge of improving access among those groups of society most in need of such programmes, for example people with lower socioeconomic status [23] and from ethnic minority communities.

### 4.3. Strengths and limitations

Our study has several strengths. First, this is a real-world evaluation of a freely available digital intervention supporting self-management for people with T2DM. Second, we used a large dataset combining the Healthy Living datasets and the latest NDA, which contains real-world data for more than 3 million people with T2DM in England. Third, using a matched control group, we evaluated intervention-related changes in clinically relevant risk factors and outcomes that predict risk for developing serious long-term diabetes-related complications. Fourth, methodologically robust data analyses were

 

employed, including multiple imputation and sequential propensity score matching to deal with confounding and to ensure robustness and the clinical relevance of the findings. Fifth, several sub-group and sensitivity analyses were performed and demonstrated the robustness of the findings reported in the main analysis, despite some caveats outlined below. Nevertheless, the study had a few limitations that may necessitate cautious interpretation of some of the reported findings. First, our follow-up period was only 1 year; longer follow-up would be desirable to assess if treatment-related differences are maintained. Second, the NDA data coverage was until March 2023 which was six months shorter than the Healthy Living registrations up to September 2023, and so to ensure participants had up to 1 year of follow-up NDA data, we restricted the Healthy Living cohort in the effectiveness analyses to participants registered up to March 2022, which reduced the sample size to only 25% of the total Healthy Living cohort. Third, the COVID-19 pandemic caused delays to general practice referral routes to Healthy Living [36], leading to lower than expected uptake of the service, and hence the diversity of registrants, in the first months of rollout of the programme. Fourth, some sensitivity and sub-group analyses were limited in power. Fifth, while we addressed confounding by controlling for measured characteristics, there may be unmeasured variations between patients which account for the differences in outcomes between groups. Sixth, no data was available to assess the impact of medication intensification on the observed effectiveness findings. Finally, the datasets lacked information on diabetes-related complications which are important outcomes in people with diabetes.

### 4.4. Potential clinical and research implications

Our findings showed modest improvements in HbA1c levels and other clinical outcomes at 1-year of Healthy Living account activation. At a population-level, the potential clinical implications are anticipated to be modest reductions in incident diabetes-related microvascular and macrovascular complications. Additionally, the intervention has potential to reduce clinical workload in primary care settings by providing patients with valuable information about diabetes. Our findings may help inform the development of new interventions to foster uptake and engagement, addressing any inequalities between population groups in DSMES programmes. Future research may use larger cohorts with recent Healthy Living data that are better aligned with the NDA, and possibly linked to hospitalisation data to assess further diabetes-related outcomes over a longer period. Further research may also identify predictors of variation in access to the intervention and similar DSMES programmes.

## 5. Conclusions

People who activated a Healthy Living account experienced, on average, modest but statistically significant health benefits compared with non-participants, and the benefits were greater in those with higher adherence. At population level these benefits would be expected to translate into modest reductions in risks for developing incident diabetes-related complications.

## Supporting information

**S1 Table. Baseline characteristics of the study groups.**
(PDF)

**S2 Table. Logistic regression model for 'Attendance' using total time spent on the website (total duration in minutes) measure as binary outcome of total time being ≤ median of 6.85 minutes (reference group) vs. > median time (N = 21,440 HL participants with usage data).**
(PDF)

**S3 Table. Logistic regression model for 'Attendance' using visits frequency measure as binary outcome of session number = 1 session (reference group) vs. > 1 session. Defined as the total number of sessions on the website visited per participant throughout the programme (N = 21,445 HL participants with usage data).**
(PDF)

**S4 Table. Baseline characteristics of the 1:5 matched HL cases-controls cohort study Groups 1 and 3.**
(PDF)

**S5 Table. Baseline characteristics of the 1:5 matched HL cases-controls cohort- groups 1 & 4 and groups 1 and 5.**
(PDF)

**S6 Table. Linear regression model output for 1-year HbA1c (mmol/mol) – primary outcome (N = 29,625).**
(PDF)

**S7 Table. Multivariable-adjusted difference (95% CI) in 1-year outcomes in HL account activators (cases) compared with matched NDA controls, excluding covariates with more than 5 or 10 percent imbalances between cases and controls. (N = 29,625).**
(PDF)

**S8 Table. Multivariable-adjusted difference (95% CI) in 1-year HbA1c model limited to participants with complete non-missing data Using matched HL activators and NDA controls cohort) (complete case analysis, CCA).**
(PDF)

**S9 Table. Multivariable-adjusted difference (95% CI) in 1-year outcomes based on different imputations (Using matched HL activators and NDA controls cohort).**
(PDF)

**S10 Table. Multivariable-adjusted difference (95% CI) in 1-year outcomes in HL account activators, HL attendees and HL completers compared with NDA controls.**
(PDF)

**S11 Table. Multivariable-adjusted difference (95% CI) in 1-year outcomes in matched HL cases and NDA controls across different HL development phases.**
(PDF)

**S12 Table. Comparison of results of selected outcomes with results from the HeLP-Diabetes trial.**
(PDF)

**S1 Fig. Baseline predictors of HL retention (attendees) in comparison to HL activators (odds ratios, 95%CI).**
(PDF)

## Acknowledgments

The authors would like to thank NHS England and the service provider for facilitating this study at all stages, e.g., by collecting and providing the data. The authors would like to thank the Healthy Living Diabetes-Long-term Independent National Evaluation (HED-LINE) advisory group and the HED-LINE Patient and Public Involvement and Engagement (PPIE) group for providing feedback and interpretations of these findings during the data analysis stages. Particular thanks to the Healthy Living programme team at NHS England, who provided valuable comments on the emerging findings during the study.

## Author contributions

**Conceptualization:** Salwa S. Zghebi, Sarah Cotterill, Martin K. Rutter, Evangelos Kontopantelis.

**Data curation:** Salwa S. Zghebi.

**Formal analysis:** Salwa S. Zghebi.

**Funding acquisition:** Sarah Cotterill, Rachel A Elliott, Brian Mcmillan, Martin K. Rutter, Evangelos Kontopantelis.

**Methodology:** Salwa S. Zghebi, Sarah Cotterill, Evangelos Kontopantelis.

**Project administration:** Salwa S. Zghebi, Sarah Cotterill.

**Supervision:** Sarah Cotterill, Evangelos Kontopantelis.

**Writing – original draft:** Salwa S. Zghebi.

**Writing – review & editing:** Salwa S. Zghebi, Sarah Cotterill, Antonia Marsden, Luke Paterson, Rachel A Elliott, Brian Mcmillan, Martin K. Rutter, Evangelos Kontopantelis.

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
