## [Decision Letter · Decision Letter 0]

19 Jan 2026

PONE-D-25-33501Examining the uptake, retention, and effectiveness of a national online type 2 diabetes self-management intervention in England (Healthy Living): a retrospective cohort studyPLOS One

Dear Dr.  Zghebi,

Thank you for submitting your manuscript to PLOS ONE. After careful consideration, we feel that it has merit but does not fully meet PLOS ONE’s publication criteria as it currently stands. Therefore, we invite you to submit a revised version of the manuscript that addresses the points raised during the review process.

**ACADEMIC EDITOR:**

While the manuscript exhibits intriguing potential, it requires substantial revisions and further refinement.<o:p></o:p>

Although the inherent interest in the subject matter is acknowledged, the reviewers have raised crucial concerns that must be properly addressed.

We look forward to receiving your revised manuscript.

Kind regards,

Marcelo Arruda Nakazone, M.D., Ph.D.

Academic Editor

PLOS One

Journal Requirements:

“MKR reports receiving consulting fees from Eli Lilly and modest GSK stock ownership both unrelated to this work. He has also led research at the University of Manchester with Innovate UK and NIHR funding to independently evaluate My Diabetes My Way, a digital intervention supporting the management of diabetes. Other authors declare no conflicts of interest.”

We note that one or more of the authors are employed by a commercial company: GSK

“This paper reports independent research funded by the National Institute for Health and Care Research (Policy Research Programme, HED-LINE, NIHR200933).”

4. We noted in your submission details that a portion of your manuscript may have been presented or published elsewhere. “Yes: 2 text-only abstracts briefly outlining the results were submitted to academic conferences” Please clarify whether this [conference proceeding or publication] was peer-reviewed and formally published. If this work was previously peer-reviewed and published, in the cover letter please provide the reason that this work does not constitute dual publication and should be included in the current manuscript.

5. Please note that your Data Availability Statement is currently missing the DOI/accession number of each dataset OR a direct link to access each database. If your manuscript is accepted for publication, you will be asked to provide these details on a very short timeline. We therefore suggest that you provide this information now, though we will not hold up the peer review process if you are unable.

Reviewer's Responses to Questions

**Comments to the Author**

1. Is the manuscript technically sound, and do the data support the conclusions?

Reviewer #1: Yes

Reviewer #2: Yes

2. Has the statistical analysis been performed appropriately and rigorously? 

Reviewer #1: Yes

Reviewer #2: Yes

3. Have the authors made all data underlying the findings in their manuscript fully available?

Reviewer #1: Yes

Reviewer #2: No

4. Is the manuscript presented in an intelligible fashion and written in standard English?

Reviewer #1: Yes

Reviewer #2: Yes

5. Review Comments to the Author

Reviewer #1: Abstract (results):

Can the authors indicate which values are crude ORs and the 95%CIs through the results for consistency?

Main manuscript

Line 143: Delete fullstop (.)

Line 186: Do you mean values or variables?

If variables, can you explain this was done beyond the stated STATA command used?

Line 221: For consistency, can all 95%CIs be presented in this style?

Line 225: From which table comes this results?

Line 232: From which table or figure comes this results, i.e. line 232-240?

Line 233: Please add the IQR.

All the tables need to be checked for:

1. Most percentages are not up to 100% or more than 100%.

Please check the whole table

2. Some figures that needed percentages were not added (e.g. Table S1, Ethnicity - Mixed, and Other)

These apply to most percentages.

Reviewer #2: In this paper, the authors reported effectiveness of online intervention in England (Healthy Living; HL), as compared to the matched-controls (NDA). The online intervention resulted in modest improvements in several metabolic parameters after 1 year. The results are almost clearly presented, and the analyses seemed to be well performed. The reviewer’s comments are as follows.

1. The contents of Healthy Living are not clearly described. Some supplementary information would be helpful.

2. Please add the description regarding ethical approval by the institution, in the main text.

3. The participants in HL program and the matched-controls (NDA) are essentially different. The participants in HL program were younger with shorter duration of diabetes and with lower past history of ischemic heart disease (Table S1). The participants in HL program might have more positive attitude to improve their glucose control. Also, there are no consideration on treatment intensification during the period. Therefore, much more caveats should be added on the improvements in metabolic parameters.

4. The improvement in HbA1c levels may be smaller than that anticipated. Some more discussion should be made regarding this observation.

5. Results, Figure 1. Were the participants who attended the program (n=8375) included those with HL account (n=21820)? If so, revise Figure 1 to avoid confusion.

6. PLOS authors have the option to publish the peer review history of their article (what does this mean?). If published, this will include your full peer review and any attached files.

Reviewer #1: **Yes:** Clement Tetteh Narh

Reviewer #2: No

---

## [Author Response · Author response to Decision Letter 1]

12 Feb 2026

Dear Editor,

Re. PONE-D-25-33501 - Examining the uptake, retention, and effectiveness of a national online type 2 diabetes self-management intervention in England (Healthy Living): a retrospective cohort study

Thank you for the opportunity to submit a revision of our manuscript.

Below, we provide a point-by-point response to the received comments. To facilitate the review process, all changes are highlighted in the revised manuscript and numbered by the Reviewer and comment numbers. For example, “Response to Reviewer 1, comment #1”.

We hope to have satisfactorily responded to the comments, which have strengthened our manuscript.

Looking forward to receiving your decision.

Yours sincerely,

Corresponding Author, on behalf of the authors

Academic Editor

Journal Requirements:

I. Please ensure that your manuscript meets PLOS ONE's style requirements, including those for file naming. The PLOS ONE style templates can be found at

https://journals.plos.org/plosone/s/file?id=wjVg/PLOSOne_formatting_sample_main_body.pdf [journals.plos.org] and

https://journals.plos.org/plosone/s/file?id=ba62/PLOSOne_formatting_sample_title_authors_affiliations.pdf [track.editorialmanager.com]

Authors response

We reviewed the manuscript and confirm that it meets the PLOS ONE’s style requirements.

II. Thank you for stating the following in the Competing Interests section:

“MKR reports receiving consulting fees from Eli Lilly and modest GSK stock ownership both unrelated to this work. He has also led research at the University of Manchester with Innovate UK and NIHR funding to independently evaluate My Diabetes My Way, a digital intervention supporting the management of diabetes. Other authors declare no conflicts of interest.”

We note that one or more of the authors are employed by a commercial company: GSK

Authors response

We confirm that the commercial affiliation did not play any role in this study. The Funding Statement has been revised to declare this commercial affiliation and the proposed statement has been also included. The amended Funding Statement now reads as follows in the manuscript:

“This paper reports independent research funded by the National Institute for Health and Care Research (Policy Research Programme, HED-LINE, NIHR200933). The views expressed in this publication are those of the authors and not necessarily those of the National Institute for Health and Care Research or the Department of Health and Social Care. The funder did not have any role in the study design, data collection and analysis, decision to publish, or preparation of the manuscript. Eli Lilly provided support in the form of consulting fees for MKR, unrelated to this work, and did not have any role in the study design, data collection and analysis, decision to publish, or preparation of the manuscript. The specific role of MKR is articulated in the ‘author contributions’ section. EK is part-funded by the NIHR HeatlhTech Research Centre in Emergency and Acute Care (NIHR205301) and the Manchester British Heart Foundation (BHF) Centre for Research Excellence (RE/24/130017).”

Within your Competing Interests Statement, please confirm that this commercial affiliation does not alter your adherence to all PLOS ONE policies on sharing data and materials by including the following statement: "This does not alter our adherence to PLOS ONE policies on sharing data and materials.” (as detailed online in our guide for authors http://journals.plos.org/plosone/s/competing-interests [journals.plos.org]) . If this adherence statement is not accurate and there are restrictions on sharing of data and/or materials, please state these. Please note that we cannot proceed with consideration of your article until this information has been declared.

Authors response

Both updated Funding Statement and Competing Interests Statement has been included in the cover letter. The proposed statement has been added to the Competing Interests Statement, and it now reads as:

“MKR reports receiving consulting fees from Eli Lilly and modest GSK stock ownership both unrelated to this work. He has also led research at the University of Manchester with Innovate UK and NIHR funding to independently evaluate My Diabetes My Way, a digital intervention supporting the management of diabetes. This does not alter our adherence to PLOS ONE policies on sharing data. Other authors declare no conflicts of interest.”

“This paper reports independent research funded by the National Institute for Health and Care Research (Policy Research Programme, HED-LINE, NIHR200933).”

Authors response

The proposed statement has been added to the Funding Statement, added to the cover letter and it now reads as:

“This paper reports independent research funded by the National Institute for Health and Care Research (Policy Research Programme, HED-LINE, NIHR200933). The views expressed in this publication are those of the authors and not necessarily those of the National Institute for Health and Care Research or the Department of Health and Social Care. The funder did not have any role in the study design, data collection and analysis, decision to publish, or preparation of the manuscript. Eli Lilly provided support in the form of consulting fees for MKR, unrelated to this work, and did not have any role in the study design, data collection and analysis, decision to publish, or preparation of the manuscript. The specific role of MKR is articulated in the ‘author contributions’ section. EK is part-funded by the NIHR HeatlhTech Research Centre in Emergency and Acute Care (NIHR205301) and the Manchester British Heart Foundation (BHF) Centre for Research Excellence (RE/24/130017).”

4. We noted in your submission details that a portion of your manuscript may have been presented or published elsewhere. “Yes: 2 text-only abstracts briefly outlining the results were submitted to academic conferences” Please clarify whether this [conference proceeding or publication] was peer-reviewed and formally published. If this work was previously peer-reviewed and published, in the cover letter please provide the reason that this work does not constitute dual publication and should be included in the current manuscript.

Authors response

Thank you for this comment. We have explained how this work does not constitute dual publication in the cover letter. Both abstracts were peer-reviewed by internal conference committees, but only one abstract was published in Diabetic Medicine supplementary issue (Diabetes UK Conference, 2025). Each abstract contains a succinct text-only summary, without detailed methodology, full results, figures/tables. The current manuscript submitted to PLOS ONE includes substantial new content, including full methods, analyses, results, and conclusions, which have never been published elsewhere. No copyright was transferred as part of the conference abstract publication. For transparency, we have fully disclosed the submission of abstracts, and we confirm that this manuscript submission does not constitute duplicate or prior publication under PLOS ONE’s policies or ICMJE Recommendations.

5. Please note that your Data Availability Statement is currently missing the DOI/accession number of each dataset OR a direct link to access each database. If your manuscript is accepted for publication, you will be asked to provide these details on a very short timeline. We therefore suggest that you provide this information now, though we will not hold up the peer review process if you are unable.

Authors response

The data underlying the results presented in the study are only available through the data provider ‘NHS England’ and authors were not provided with and DOI/accession number. All interested researchers can request to access the data through NHS England directly and authors are not permitted to share the data publicly (even as minimal underlying dataset) or make it available as per the data use agreement with NHS England. The authors did not have any special access privileges to this data.

Authors response

The following ethics statement is now included in the Methods section of the manuscript as follows: “This study was reviewed and approved by the Yorkshire & The Humber - Leeds West Research Ethics Committee on 18 August 2020 (reference number: 20/YH/0250). The research team undertook secondary analysis of pseudonymised datasets and were not responsible for data collection or obtaining informed consent.”

7. Please include captions for your Supporting Information files at the end of your manuscript, and update any in-text citations to match accordingly. Please see our Supporting Information guidelines for more information: http://journals.plos.org/plosone/s/supporting-information [journals.plos.org].

Authors response

Captions for the Supporting Information files are now added at the end of the manuscript, and in-text citations have also been reviewed and updated.

Authors response

The reviewers' comments did not recommend including additional references; hence no new citations were added to the manuscript.

Reviewer #1

Abstract (results):

1) Can the authors indicate which values are crude ORs and the 95%CIs through the results for consistency?

Authors response

Thank you for this comment. All reported ORs are estimated using multivariable logistic regression models i.e. adjusted ORs. We did not feel univariable associations would add any value to the paper, and any such estimates were not directly relevant to our research question. We have now corrected this in the abstract to indicate that all reported ORS are adjusted ORs.

Changes to the paper

• Abstract: Removed any mention of ‘aOR’ from the abstract and all are now reported as 'OR' for consistency.

• Methods section (Line 170): the relevant text now reads as “Using multivariable logistic regression models, we estimated adjusted odds ratios (OR) and 95% confidence intervals (95% CI)…”

Main manuscript:

2) Line 143: Delete fullstop (.)

Authors response

The full stop has been deleted.

Changes to the paper

The text now reads as: "2.4.1. Uptake and retention of Healthy Living"

3) Line 186: Do you mean values or variables? If variables, can you explain this was done beyond the stated STATA command used?

Authors response

We thank the reviewer for this comment. We meant values of baseline variables and have revised the text for clarity.

Changes to the paper

The text now reads as (Line 195): "Multiple imputation using the mi suite of commands in Stata was used to impute missing values of baseline variables and outcomes (10 imputations)."

4) Line 221: For consistency, can all 95%CIs be presented in this style?

Authors response

All 95% CIs have been revised and presented as per the suggested style throughout the paper.

5) Line 225: From which table comes this results?

Authors response and Changes to the paper

We would clarify to the reviewer that the reported results of the retention analysis are presented in Figure S1 which is now cited earlier in the text for clarity (Line 253).

6) Line 232: From which table or figure comes this results, i.e. line 232-240?

Authors response

Thank you for this comment and we apologise for the omission.

Changes to the paper

In response to this comment, the full tables have now been added to the supplementary file as Tables S2 and S3 and cited in the main manuscript as follows (Line 264-267):

“We compared longer attendees, who attended for more than the median time (N=10,710), and shorter attendees, who attended for the median or less (N=10,730). Longer attendees were, on average less likely to be male or to be of Asian or Black ethnicity, less likely to be prescribed insulin or non-insulin diabetes medication or statins, and their time since T2DM diagnosis was lower than shorter attendees (S2 Table). People who attended multiple sessions, were on average older, less likely to be male or to be of Asian ethnicity, to be from the most deprived areas, prescribed non-insulin diabetes medication or statins, and their time since T2DM diagnosis was shorter when compared with those who attended one session (S3 Table).”

7) Line 233: Please add the IQR.

Authors response and Changes to the paper

Thank you for this comment. The IQR has now been added to the manuscript and reads as (Line 260):

“….the median time spent on Healthy Living website was 6.9 mins (IQR 0.4 - 26.3).”

All the tables need to be checked for:

8) Most percentages are not up to 100% or more than 100%.

Please check the whole table

Authors response

We would like to highlight that we have been directed to apply the data provider's Statistical Disclosure Control rules to all reported figures which involved: ‘Small Number Suppression’ and ‘Rounding’ figures to the nearest five to prevent identification. As a result, it is standard for totals and percentages to not sum exactly, and we confirm that it does not indicate an error in the analysis or reporting.

Changes to the paper

We have added the following text in the Methods section and as a footnote to all tables to indicate the reason of any apparent total/percentage discrepancies, and it reads as follows:

“In accordance with mandatory data provider Statistical Disclosure Control (SDC) rules (such as, rounding and small number suppression), individual categories may not sum to the total, and percentages may not sum to 100%.”

9) Some figures t

---

## [Decision Letter · Decision Letter 1]

14 Apr 2026

Examining the uptake, retention, and effectiveness of a national online type 2 diabetes self-management intervention in England (Healthy Living): a retrospective cohort study

PONE-D-25-33501R1

Dear Dr. Zghebi,

We’re pleased to inform you that your manuscript has been judged scientifically suitable for publication and will be formally accepted for publication once it meets all outstanding technical requirements.

Kind regards,

Marcelo Arruda Nakazone, M.D., Ph.D.

Academic Editor

PLOS One

Additional Editor Comments (optional):

Reviewers' comments:

Reviewer's Responses to Questions

**Comments to the Author**

1. If the authors have adequately addressed your comments raised in a previous round of review and you feel that this manuscript is now acceptable for publication, you may indicate that here to bypass the “Comments to the Author” section, enter your conflict of interest statement in the “Confidential to Editor” section, and submit your "Accept" recommendation.

Reviewer #2: All comments have been addressed

2. Is the manuscript technically sound, and do the data support the conclusions?

Reviewer #2: Yes

3. Has the statistical analysis been performed appropriately and rigorously? 

Reviewer #2: Yes

4. Have the authors made all data underlying the findings in their manuscript fully available?

Reviewer #2: Yes

5. Is the manuscript presented in an intelligible fashion and written in standard English?

Reviewer #2: Yes

6. Review Comments to the Author

Reviewer #2: Thank you for your rigorous revision. This reviewer thinks that the paper is now found to be acceptable.

7. PLOS authors have the option to publish the peer review history of their article (what does this mean?). If published, this will include your full peer review and any attached files.

Reviewer #2: No

---

## [Editor Report · Acceptance letter]

PONE-D-25-33501R1

PLOS One

Dear Dr. Zghebi,

I'm pleased to inform you that your manuscript has been deemed suitable for publication in PLOS One. Congratulations! Your manuscript is now being handed over to our production team.

Kind regards,

on behalf of

Professor Marcelo Arruda Nakazone

Academic Editor

PLOS One